**DOI: 10.1038/ncomms12448**　　**OPEN**

# Blue-sky bifurcation of ion energies and the limits of neutral-gas sympathetic cooling of trapped ions

Steven J. Schowalter[1], Alexander J. Dunning[1], Kuang Chen[1], Prateek Puri[1], Christian Schneider[1] & Eric R. Hudson[1]

Sympathetic cooling of trapped ions through collisions with neutral buffer gases is critical to a variety of modern scientific fields, including fundamental chemistry, mass spectrometry, nuclear and particle physics, and atomic and molecular physics. Despite its widespread use over four decades, there remain open questions regarding its fundamental limitations. To probe these limits, here we examine the steady-state evolution of up to 10 barium ions immersed in a gas of three-million laser-cooled calcium atoms. We observe and explain the emergence of nonequilibrium behaviour as evidenced by bifurcations in the ion steady-state temperature, parameterized by ion number. We show that this behaviour leads to the limitations in creating and maintaining translationally cold samples of trapped ions using neutral-gas sympathetic cooling. These results may provide a route to studying non-equilibrium thermodynamics at the atomic level.

[1] Department of Physics and Astronomy, University of California, Los Angeles, California 90095, USA. Correspondence and requests for materials should be addressed to S.J.S. (email: schowalt@physics.ucla.edu).

The concept of using a background gas to facilitate the loading of metallic particles into early ion traps was first demonstrated in 1959 (ref. 1). Since then, using collisions with cold, chemically inert gases to sympathetically cool trapped ions has become a general technique in many areas of scientific research. Today, the sympathetic cooling of ions with neutral buffer gases is routinely used to control the energy scales of chemical reactions[2], to improve the performance of commercial mass spectrometers[3], and to prepare short-lived exotic nuclei for tests of the Standard Model[4]. It has also been proposed as a method to initialize qubits in future quantum computation architecture[5].

Despite the widespread use of sympathetic cooling in ion traps, its kinetics have not yet been completely described and, likewise, its limitations are still not fully understood. The complexity of the cooling process derives from the fact that a system of trapped ions immersed in a cold buffer gas is not an isolated system. Instead, due to the presence of the time-dependent trapping potential, energy is injected and removed from the system over a single trapping cycle. Thus, as it is not a true representation of the canonical ensemble, the trapped ions do not tend to thermal equilibrium with the cold buffer gas as one might expect. As a result, collisions in the ion trap lead to fundamentally nonequilibrium processes that have made it difficult to establish a complete kinetic description of the technique and, as later explained, limits its ability to create and maintain translationally cold temperatures.

To probe this nonequilibrium behaviour, we characterize the collisional processes between trapped ions and a buffer gas of laser-cooled atoms in a hybrid atom–ion trap. In our conception of this apparatus (shown in Fig. 1a), laser-cooled barium ions confined by a linear quadrupole trap (LQT) are immersed in a 4 mK gas of magneto-optically trapped calcium atoms. Thus far, relatively little work has been done to fully understand the complex statistical mechanics of these hybrid systems[6–11], despite being critical to current applications such as observing atom–ion collisions and reactions at cold temperatures[7,12–16] and

producing cold molecular ions[17,18]. Through a more complete understanding of the collisional processes and nonequilibrium phenomena in these systems, which offer precise experimental control, we can establish the general limits of sympathetic cooling in ion traps.

In what follows, we first present a full description of the sympathetic collisional processes in a hybrid trap and how they lead to the emergence of nonequilibrium behaviour. Second, we provide the results of a molecular dynamics (MD) simulation that confirms the emergence of nonequilibrium dynamics of small numbers of trapped ions in a laser-cooled buffer gas under idealized conditions. Third, we describe an experiment in which we immerse laser-cooled barium ions in a gas of ultracold calcium atoms and observe the hallmark of this nonequilibrium behaviour: bifurcation of the ion steady-state temperature as parameterized by trapped ion number. Last, we remark on what these findings entail regarding the limits of sympathetic cooling of trapped ions with neutral buffer gases, we discuss the implications of the results on current hybrid trap experiments, and we expound on ways to engineer future hybrid traps to tailor more diverse sources of heating and cooling to model a variety of nonequilibrium systems.

## Results

**Analytical model of the sympathetic cooling process.** Since the work of Major and Dehmelt[19], it has been known that even a single trapped ion immersed in a buffer gas does not equilibrate to the temperature of the buffer gas. Recent work has explained that this is due to a phenomenon, termed 'micromotion interruption'[10,11,20], whereby collisions with buffer gas particles disrupt the ion micromotion and couple energy from the time-dependent trapping potential into the kinetic energy of the recoiling ion. This interruption leads to multiplicative noise in the fluctuation spectrum of the trapped ions and, in turn, to non-Maxwellian statistics and power-law tails in the distributions of the steady-state ion energy[6,11]. For this reason, when referring to

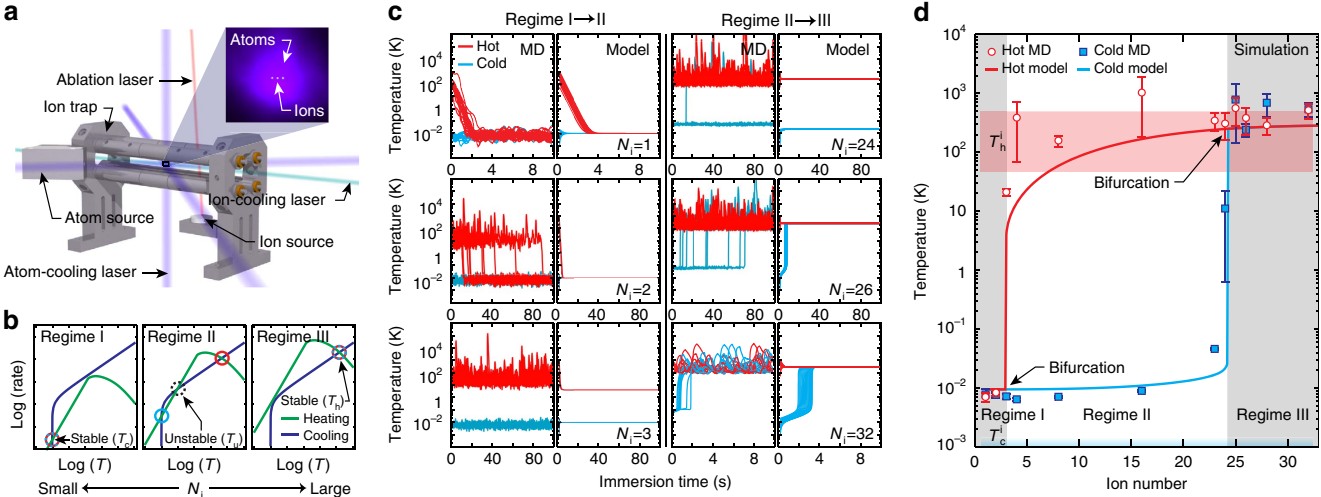

**Figure 1 | Emergence of nonequilibrium dynamics in a hybrid atom–ion trap.** (**a**) A 3D rendering of the hybrid trap used to create an ensemble of trapped ions immersed in a laser-cooled buffer gas, as shown in the inset composite photograph. (**b**) Analytical heating and cooling rates for ions in a hybrid atom–ion trap. Three characteristic regimes exist for different relative heating and cooling strengths as parameterized by $N_i$ and are characterized by the number and type of fixed points they exhibit. (**c**) Steady-state evolution of ion temperatures for different $N_i$ and initializations as calculated by the MD simulation and analytical model. The left (right) pairs convey this evolution near the opening (closing) of the bifurcation at $N_i = 3$ ($N_i = 26$), where the number of steady-state temperatures go from one (two) to two (one). (**d**) Steady-state temperatures for different $N_i$ and initializations as calculated by the MD simulation and analytical model. Error bars represent s.e. Red (blue)-shaded regions represent the ranges of ion temperatures before immersion for the hot (cold) initialization. Regimes I, II, and III are observed and indicated by differently shaded regions. The transitions between these regimes are marked by blue-sky bifurcations.

the 'temperature', $T$, of the ions, we are in fact referring to the total kinetic energy $W$ per ion given from $W/N_i = \frac{3}{2}k_B T$ with $N_i$ being the number of ions and $k_B$ the Boltzmann constant. Considering the collisional effects of micromotion interruption, ref. 11 has shown that the temperature of a single ion immersed in a buffer gas at temperature $T_n$ decreases exponentially towards a steady-state temperature $\overline{T_n} \geq T_n$, according to the rate

$$Q_{i-n} = -\Gamma_l \lambda (T - \overline{T_n}) \qquad (1)$$

where $\Gamma_l = 2\pi\rho_n\sqrt{\frac{\alpha_n}{2\mu}}$ is the Langevin collision rate, with $\rho_n$ being the buffer gas density, $\alpha_n$ the cold-atom polarizability and $\mu$ the atom–ion reduced mass. The kinetics factor $\lambda$—the smallest eigenvalue of a three-dimenional relaxation matrix[11]—depends on the atom–ion mass ratio $\tilde{m}$ and a critical mass ratio $\tilde{m}_c$, above which, interestingly, the buffer gas heats the ion regardless of $(T - \overline{T_n})$; for the hybrid system discussed here, $\tilde{m} < \tilde{m}_c$. The absolute value of this cooling rate and its dependence on $T$ is shown in Fig. 1b. This model assumes an isotropic cold-atom reservoir, isotropic momentum transfer differential cross-section and does not incorporate short-range dynamics due to the $C_4$ atom–ion interaction[10].

In addition to this cooling rate, an effective heating rate is introduced when multiple ions are present in the trap, which arises from the mutually repulsive Coulomb force between ions. Simply put, while a single ion in a LQT exhibits a stable Mathieu trajectory with a conserved average energy, co-trapping of additional ions leads to trajectories whose kinetic energy is not conserved, but grows as work is done on the ions by the time-dependent trapping field. This heating rate is simply the product of an effective ion–ion collision rate $\Gamma_{i-i}$ and the fractional increase in ion energy per collision $\bar{\epsilon}$, which vanishes for static traps[21], but is nonzero for LQTs. $\Gamma_{i-i}$ is traditionally approximated using the Chandrasekhar-Spitzer plasma self-collision rate[22], while the determination of $\bar{\epsilon}$ is given in ref. 20. Ultimately, this heating rate can be approximated by

$$Q_{i-i} = \frac{e^4\rho_i\log\Lambda}{2\pi\epsilon_0^2\sqrt{m_i}(3k_B\eta T)^{3/2}}\bar{\epsilon}T\xi(N_i) \qquad (2)$$

with $e$ being the electron charge, $\epsilon_0$ the permittivity of free space, $\log\Lambda$ the Coulomb logarithm, $\rho_i$ the ion density, $m_i$ the ion mass and $\eta$ the ratio of secular to total kinetic energy (see Methods). The inclusion of the empirically motivated function $\xi(N_i)$ extends the applicability of this model from large, three-dimensional ion clouds to small, linear Coulomb crystals[23] (see Methods). For fixed $N_i$, this heating rate displays peculiar behaviour as a function of temperature, decreasing at both high and low temperatures due to the concomitant reduction of ion density and highly-correlated ion motion, respectively. However, at intermediate temperatures, the ions exist in a liquid-like state where they are closely spaced and strongly interacting, yet not sufficiently cold to form an ordered structure, resulting in a maximum heating rate as shown in Fig. 1b.

By combining these treatments of heating and cooling in a hybrid trap, we arrive at an analytical model that describes the temperature of multiple trapped ions immersed in a buffer gas. Qualitatively, three regimes exist and are shown in Fig. 1b. Regime I, which is characterized by small ion number, features only one temperature, $T_c$, for which the heating and cooling rates are balanced, implying the existence of a unique steady-state for that ion number. In regime II, a larger ion number shifts the heating curve upwards, creating two additional intersections between the heating and cooling rates at $T_u$ and $T_h$. Ions initialized with a temperature greater (less) than the unstable fixed point at temperature $T_u$ will evolve to the steady-state temperature $T_h$ ($T_c$). In regime III, an overabundance of ions

causes the heating rate to overwhelm the cooling rate leaving only one intersection and steady-state temperature $T_h$. The boundaries of these regimes are marked by blue-sky, or saddle-node, bifurcations[24], where a second steady-state temperature is either created (I→II) or annihilated (II→III) simply by increasing the number of trapped ions by one.

**MD simulation.** In addition to this analytical model, we perform a MD simulation in which ions interact with a time-dependent trapping potential, other co-trapped ions, and a homogeneous buffer gas of 4 mK atoms with a density of $2 \times 10^{10}\,\mathrm{cm}^{-3}$ (see Methods). The MD result naturally includes effects beyond the second moment of the distribution and therefore serves to verify the analytical results at these small ion numbers. The bifurcation of the ion steady-state temperature is probed by initializing the ions to different temperatures, $T_h^i$ and $T_c^i$, and monitoring the evolution of the ion temperature after immersion in the laser-cooled buffer gas. For cold initializations, $T_c^i \leq 1$ mK, as indicated in Fig. 1d by the the blue-shaded region, and for hot initializations, $50\,\mathrm{K} \leq T_h^i \leq 500\,\mathrm{K}$, as indicated by the red-shaded region.

Figure 1c shows the results of 10 MD trials of both hot and cold initializations for six different ion numbers with immersion times up to 100 s. For comparison, the results from the analytical model using identical parameters and a similar range of initial hot and cold temperatures are shown to the right of each MD result. The left MD model pairings in Fig. 1c convey the ion kinetics near the first bifurcation at the transition between regimes I and II. For $N_i = 1$, both hot and cold initializations quickly tend to the same steady-state temperature $T_c \sim 10$ mK in roughly 2 s and agree well with the model. For $N_i = 2$, the system remains in regime I, yet the time required for all hot MD trials to reach the cold steady-state temperature is considerably longer ($\sim 100$ s). This is due to there being only a subset of configurational phase space in which two ions are able to be cooled by the laser-cooled buffer gas. As the hot ion pair oscillates in the trap, phase space is sampled until a configuration amenable to this cooling is reached. At this point, assuming no further cooling-inhibiting rearrangements occur, the ion temperature decreases with a time constant similar to that for the single-ion case. This sampling of configurational phase space introduces a probabilistic aspect to the cooling process which the analytical model does not incorporate. For $N_i = 3$, none of the hot trials exhibit cooling even after 100 s, implying that the inclusion of the additional ion renders the probability of reaching a coolable configuration vanishingly small. Instead, the existence of two bifurcated steady-state temperatures, $T_c$ and $T_h$, are revealed, which is indicative of the transition into regime II.

The right MD model pairings in Fig. 1c depict the ion kinetics near the second bifurcation at the transition between regimes II and III. For $N_i = 24$, the system appears to remain in regime II as indicated by the two unique steady-state temperatures after 100 s of immersion time. However, a single MD trial is shown to 'hop' from the cold steady state to the hot steady state, hinting at the onset of a transition to regime III. For $N_i = 26$, every cold trial abruptly hops from the once-cold steady state over a 100 s period leading to a unique steady state at $T_h$. In the small ion-number limit, heating is the result of discrete collisional events rather than the smooth rates expressed in the analytical model. Thus, the transition from regime II to III is marked by fluctuations in the ion energy that probabilistically drive the ions into configurations with temperatures above $T_u$, at which point the ion temperatures converge to the hot steady-state $T_h$. As the ion number continues to increase, this hopping happens more rapidly as a result of more frequent collisional events, as shown for $N_i = 32$ in Fig. 1c.

Finally, Fig. 1d shows the mean steady-state temperatures and corresponding s.e. of the 10 hot and cold trials for 12 different ion numbers. The hot and cold steady-state temperatures, according to the analytical model, are in good agreement with the MD result. Both results clearly identify blue-sky bifurcations near $N_i = 3$ and $N_i = 25$, demonstrating that nonequilibrium behaviour is indeed a hallmark of this hybrid system.

**Observation of emergent nonequilibrium behaviour.** To experimentally probe these bifurcations in our hybrid atom–ion trap, we immerse trapped barium ions in a gas of ultracold calcium atoms. We measure the steady-state temperature of the trapped ions as a function of both the number of trapped ions $N_i$ and initial ion temperatures, $T_h^i$ and $T_c^i$. The apparatus used here is described in Methods and elsewhere[17]. Each measurement begins with a linear Coulomb crystal of laser-cooled barium ions (in the absence of the cold calcium buffer gas) and is subsequently divided into three steps: initialization, immersion and recooling (Fig. 2). Hot initialization, demarked by the disappearance of the ion fluorescence in the second frame in Fig. 2, is performed by applying white noise to the LQT electrodes (100 ms with laser-cooling followed by 100 ms without). Cold initialization is performed simply by maintaining the laser-cooling of the ions until the beginning of immersion, as indicated by the persistence of ion fluorescence in the second frame of the cold measurement in Fig. 2.

Once initialized, immersion begins by introducing an overlapping buffer gas of 4 mK calcium atoms with a density of $2.3(5) \times 10^9$ cm$^{-3}$ and a $1-\sigma$ radius of 0.46(4) mm. Collisions between ions and atoms occur for 5 s, equivalent to $\sim 17/\Gamma_1$. Following immersion, the laser-cooled buffer gas is removed and laser-cooling of the ions is resumed. During this recooling period, the reappearance of ion fluorescence (Fig. 2) is monitored for up to 15 s. The recooling time, defined as the time needed for the $N_i$ ions to recrystallize, indicated by the marked (*) images in Fig. 2, serves as an indirect measurement of the steady-state ion temperature, as demonstrated in ref. 25). In addition to the hot and cold measurements, we perform two control measurements in which the ions are similarly initialized but the buffer gas is never present, precluding steady-state bifurcation. As seen in Fig. 2, the recooling times for the hot and cold measurements are each larger than the respective hot and cold control

measurements. For the cold measurements, this is an expected and general trend due to laser-cooling resulting in initial temperatures less than $\overline{T_n} \sim 10$ mK. For the hot measurements in regimes II and III, this is due to the fact that the hot initialization aims only to prepare the ions above $T_u$, which is less than $T_h$.

We include an additional laser to quickly dissociate calcium dimer ions that are produced through photo-associative ionization in the laser-cooled buffer gas[26]. This leads to the creation of a background of calcium ions, due to both the dissociation of the calcium dimer ions and the direct ionization of excited neutral calcium atoms. To reduce the effect of this background on the barium ions, we choose LQT parameters that render calcium ions unstable and therefore absent shortly ($\sim 3$ μs) after production. We estimate the average number of calcium ions in the trap at any given time to be $\sim 0.02$ (see Methods). This Ca$^+$ background adds a small but non-negligible heat load to the ions which can also help explain why hot and cold measurements are each larger than the respective hot and cold control measurements.

The results of the experiment are shown in Fig. 3a, where each point and error bar represent the mean recooling time and its corresponding s.e. for each $N_i$. For $2 \leq N_i \leq 6$, we observe a statistically significant difference between the recooling times of the hot and cold measurements. This difference is also reflected in the recooling time distributions for the same range of ion numbers, as shown by the top panel of Fig. 3b. Here the hot distribution is shown to be skewed towards longer recooling times relative to the cold distribution, implying a higher temperature for the hot measurement. We interpret this as an indication of the system being in regime II for $2 \leq N_i \leq 6$. As we increase the number of ions ($N_i \geq 7$), we observe statistically similar mean recooling times for both the hot and cold measurements. This similarity is also reflected by the hot and cold recooling time distributions for $N_i \geq 7$, shown in the bottom panel of Fig. 3b. We interpret this as the system completing its transition from regime II to III and evidence of a blue-sky bifurcation.

The similar recooling times for $N_i = 1$ are evidence of the sympathetic cooling power of the laser-cooled buffer gas, but technically should not be considered evidence of regime I since $Q_{i-i}(N_i = 1) = 0$. For $N_i > 1$, the experimentally achievable cold-atom densities are insufficient to access regime I—this regime is likely accessible in hybrid traps using cold alkali atoms where densities are typically 10–100 times larger. Also shown in Fig. 3a

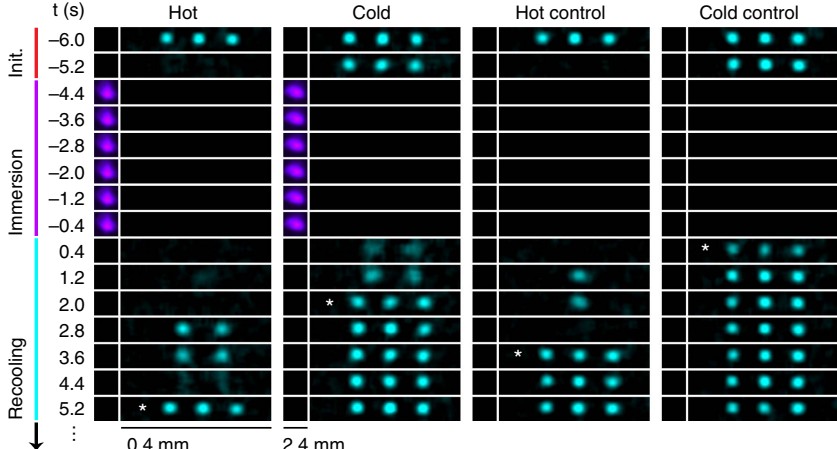

**Figure 2 | Experimental duty cycle.** Photographs of the ion (teal) and atom (purple) fluorescence during laser-cooling. Each measurement is split into initialization, immersion, and recooling. Ions are initialized either hot or cold. During immersion, the ions are immersed in the laser-cooled buffer gas for 5 s followed by a recooling period. Two control measurements are included during which the buffer gas is not present. The marker (*) indicates the frame at which recooling of the immersed ions has been achieved, which defines the recooling time for that trial.

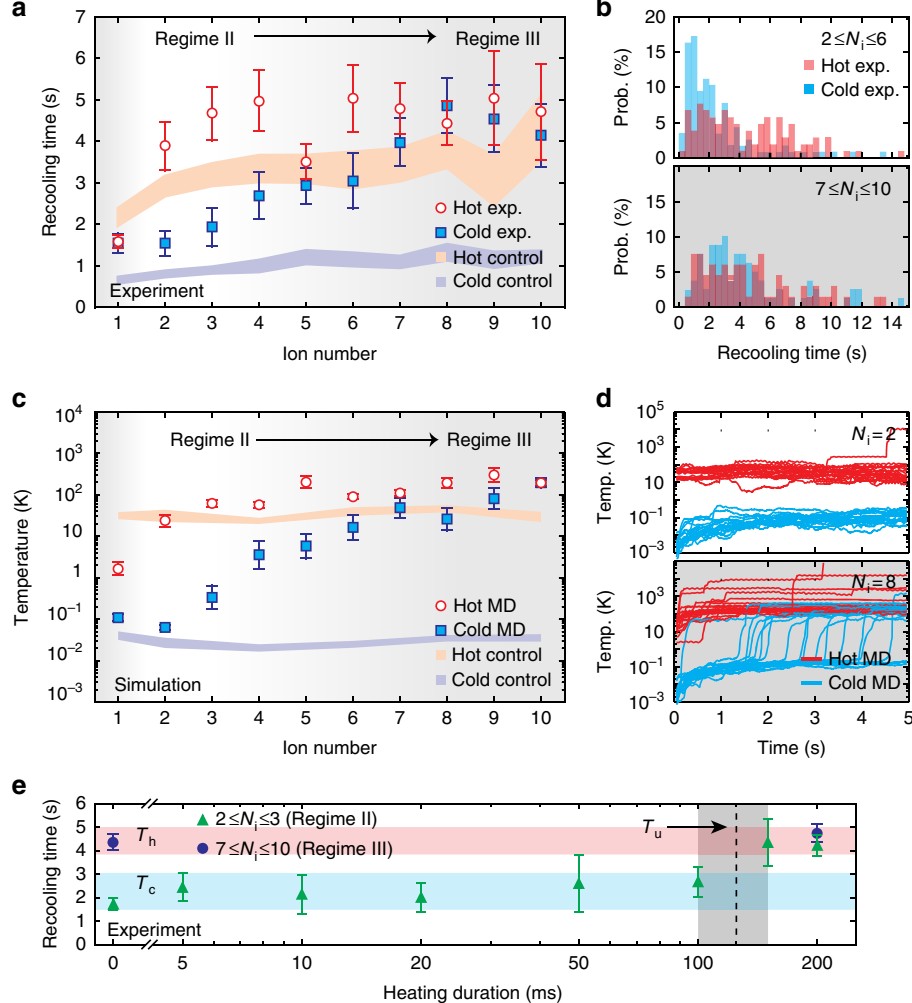

**Figure 3 | Observation of a blue-sky bifurcation in ion energy. (a)** Bifurcation diagram of the mean recooling times for $N_i = 1$–10 with error bars reflecting the s.e. For $2 \leq N_i \leq 6$, statistically different recooling times are observed (indicative of regime II), which then converge for $N_i \geq 7$ (indicative of regime III). Shaded regions represent one s.e. about the mean recooling times for the control data and do not exhibit bifurcation. **(b)** Histograms of experimental recooling times accordingly convey the difference ($2 \leq N_i \leq 6$) and similarity ($N_i \geq 7$) in the steady-state temperatures. **(c)** Bifurcation diagram of steady-state temperatures, as calculated by a MD simulation using experimental conditions, is consistent with the experimentally observed bifurcated feature shown in **a**. Error bars represent the s.e. **(d)** MD-calculated traces showing the steady-state evolution for $N_i = 2$ and 8, revealing the occurrence of steady-state 'hopping' in regime III. **(e)** Experimental observation of the bifurcation temperature $T_u$ in regime II by varying the duration of the heating during hot initialization. Below (above) 100 ms of heating the recooling times are indicative of the cold (hot) steady state shown schematically by the blue (red) shaded region, and corresponds to $T_u \sim$ 1-50 K. The grey region indicates the uncertainty associated with $T_u$ and error bars represent s.e. As expected, data from regime III does not exhibit this abrupt increase in recooling times.

are the recooling times for the hot and cold controls. As expected, without the presence of the buffer gas, no bifurcation in the recooling times is observed.

To further interpret these results, we perform a MD simulation using parameters similar to those used in the experiment which include, for example, non-uniform calcium atom density, electrical noise on the trap electrodes and production of unstable calcium ions (see Methods). The results of this simulation are shown in Fig. 3c, with each point representing the mean ion temperature and its corresponding s.e. after 5 s of immersion for 20 trials. Again, we see a similarly separated region for the same range of ion numbers which gradually closes near $N_i = 7$. We find that the gradual and premature closing of the bifurcation, and transition into regime III, is primarily due to a prevailing heating effect caused by the infrequent and transient presence of unstable calcium ions. Without this unwanted heating effect, simulations predict the prevalence of regime II beyond $N_i = 10$.

The MD temperature traces for two and eight ions are shown in Fig. 3d. The top panel ($N_i = 2$) reveals the existence of two bifurcated steady-state temperatures indicative of regime II. The bottom panel ($N_i = 8$) shows cold traces that exhibit hopping to the hot steady state as the system transitions into regime III. The gradual increase in temperature for cold simulations, illustrated in Fig. 3c, is therefore explained by the increasing probability of a steady-state changing hop occurring during the 5 s immersion period. This probability grows from virtually zero at $N_i = 2$ to near-unity at $N_i = 8$.

We perform an additional experiment to probe the mean bifurcation temperature $T_u$, defined as the initial ion temperature above (below) which the system tends to the hot (cold) steady state in regime II for $2 \leq N_i \leq 3$. Here the ions are again preinitialized at laser-cooled temperatures and subsequently heated as before, but now for a variable duration to scan the initial temperature $T_h^i$. Following this initialization, the ions are

again immersed for 5 s and then recooled. As seen in Fig. 3e, ions which are heated for $\leq 100$ ms are each recooled in 2.3(8) s, yet when the heating duration exceeds 100 ms, the recooling times abruptly increase to 4.4(6) s. These recooling times are consistent with those of the cold and hot measurements for $2 \leq N_i \leq 3$ shown in Fig. 3a, respectively. We therefore interpret this sudden increase in recooling times after 100 ms as evidence of the system hopping from the cold steady-state $T_c$ to the hot steady-state $T_h$ once $T_h^i > T_u$. With the aid of the MD simulation, we estimate $T_u$ for $2 \leq N_i \leq 3$ to be $\sim 1$–50 K. As expected for regime III, we see no evidence of the cold steady-state $T_c$.

## Discussion

The kinematics responsible for these bifurcations can have serious implications for sympathetic cooling in ion traps. The interplay between the Coulomb logarithm and the ion density results in the maximum ion–ion heating rate occurring at temperatures of a few kelvin. This means that sympathetic cooling using cryogenic buffer gases could result in ion temperatures significantly greater than that of the buffer gas. For example, using a 4 K $^4$He buffer gas with a density of $10^{14}$ cm$^{-3}$, we estimate the steady-state ion temperature for $N_i(Ba^+) \gtrsim 32$ to be $\sim 10$ K. In fact, using a 170 mK $^3$He buffer gas with the same density, we still estimate the steady-state ion temperature to be $\sim 10$ K. This illustrates the surprising diminishing returns of naively decreasing the buffer gas temperature to reach colder steady-state ion temperatures. With this in mind, the atom–ion collision energies of chemical reaction rates measured using sympathetic cooling in ion traps may need to be re-evaluated.

To overcome this limitation, high-density buffer gases composed of laser-cooled, highly-polarizable atoms, such as those created in hybrid traps, must be used (Table 1). However, even with current hybrid traps, for more than a critical number of ions, it is only possible to maintain, not create, steady-state ion temperatures near that of the laser-cooled buffer gas. For still larger ion numbers and with certain buffer gases (typical low-density alkaline-earth gases), it may not even be possible to maintain these laser-cooled temperatures (for example, for Ca). In most cases, these issues can be circumvented either by directly laser-cooling the trapped ions or by sympathetically cooling the ions with co-trapped laser-cooled ions, which can maintain (or simply initialize, in the case of cooling with Na and Rb gases) ions at sub-millikelvin temperatures.

Our studies have also found that several experimental details, such as finite size of the buffer gas and electrical noise, can have profound effects on the evolution of the ion towards steady state and perhaps could be used to further explore the rich physics of these systems. For example, the sensitivity of trapped ions to electrical noise demonstrates the interesting possibility of applying tailored time-dependent potentials to trap electrodes to study diverse work protocols and their influence on system dynamics. This opens the door for the study of arbitrarily driven nonequilibrium systems, such as extensions of molecular motors

and recently realized single-atom heat engines[27]. Furthermore, using this electrical control, it should be possible to prepare a system at different points in phase space and subsequently monitor its relaxation (or lack thereof), perhaps shedding light on ergodicity in nonequilibrium systems. In addition, future hybrid atom–ion traps can be engineered to allow for the immersion of ions in multiple non-interacting laser-cooled buffer gases. These multicomponent systems may give rise to interesting phenomena[28] such as the breakdown of the fluctuation-dissipation theorem, further departures from Maxwellian velocity distributions, and the non-factorizability of joint position-velocity probability distributions leading to position-velocity sorting.

In summary, we have presented a theoretical and experimental description of trapped ions immersed in a cold atomic gas, demonstrating the emergence of nonequilibrium behaviour, culminating in the observation of a blue-sky bifurcation in the ion steady-state temperature. The impact of this work is primarily twofold: first, the cooling power of a laser-cooled buffer gas is only sufficient to cool the translational motion of small numbers of trapped ions. This has serious implications for hybrid atom–ion traps, which are typically employed to study physics and chemistry at ultracold temperatures, and places limits on the ability to create and maintain cold samples of trapped ions using sympathetic cooling. If systems with larger numbers of ions are desired, the restricted cooling power necessitates the need for additional cooling, such as sympathetic cooling with co-trapped laser-cooled ions. Outside of hybrid trapping, these limitations may impact low-temperature chemical studies that use neutral gases to set reaction energies and the development of mass spectrometers, which use cold neutral gases to collisionally cool analyte ions for increased mass resolution. This work may also revise the viability of using cold neutral gases alone as an initialization step for molecular ion qubits in future quantum computation architecture. Additionally, the ion–ion heating rate applied in the present work should also be applicable to studies of bistability in laser-cooled trapped ions. Second, this work represents the first experimental realization of the rich non-equilibrium physics that can exist in hybrid traps. Given the precise control afforded by these systems and the ease of implementation of work protocols and dissipative mechanisms, hybrid systems may become precise tools with which to study the complex phenomena of systems existing far from equilibrium.

## Methods

**Analytical model.** The cooling rate defined in equation (1) is developed in ref. 11. The kinetics factor $\lambda$ suppresses the cooling rate which accounts for the less-than-unit cooling efficiency per collision. As referred to in the text, this inefficiency is a consequence of ongoing micromotion interruption that results in some collisions leading to cooling and others to heating. As $\lambda$ approaches zero, fewer collisions result in cooling. The micromotion interruption also leads to the cooling asymptote $\overline{T}_n$ being greater than the reservoir temperature $T_n$. The micromotion interruption is a function of trap parameters and atom masses, but generally worsens for increasing Mathieu-$q$ and $m_n/m_i$ ratio.

**Table 1 | Temperature limits.**

| Gas | $\alpha$ (a.u.) | $\rho$ (cm$^{-3}$) | $T_n$ | $T_{create}$ | $T_u$ | $T_{maintain}$ |
|---|---|---|---|---|---|---|
| $^3$He | 1.38 | $10^{14}$ | 170 mK | 10 K | 2 K | 300 mK |
| $^4$He | 1.38 | $10^{14}$ | 4 K | 10 K | — | — |
| $^{23}$Na | 163 | $10^{12}$ | 100 $\mu$K | 30 K | 800 mK | 200 $\mu$K |
| $^{87}$Rb | 319 | $10^{12}$ | 10 $\mu$K | 30 K | 800 mK | 40 $\mu$K |
| $^{40}$Ca | 157 | $10^{10}$ | 4 mK | 300 K | — | — |

Expected steady-state temperatures of Ba$^+$ ions for the large $N_i$ limit ($N_i \geq 32$) after being immersed in a variety of gases with polarizabilities $\alpha$, densities $\rho$ and temperatures $T_n$. $T_{create}$ ($T_{maintain}$) is the temperature reached if the ions are initialized above (below) $T_u$.

The heating rate defined in equation (2) is developed in ref. 20. Rather than model the secular temperature $T_{sec}$, here we approximate the total ion temperature $T$ such that $T_{sec} = \eta T$. The ratio of secular to total temperature $\eta$ is a function of trap parameters and decreases as micromotion increases relative to the secular motion. In practice, it is convenient to treat micromotion as an additional degree of freedom. Here the axially confining field is much weaker than the radially confining field that leads to micromotion being relevant in only two dimensions. Then, by equipartition, $\eta \sim \frac{3}{5}$. Exact calculation of $\eta$ using ref. 20 yields 0.50.

For small numbers of ions, $\rho_i$ is poorly defined and it becomes impractical to derive an analytical form for the ion–ion heating rate due to its dependence on the actual ion trajectories. We therefore modify the heating rate by the inclusion of the function $\xi(N_i) = 1 - e^{-(N_i/N_k)^3}$. This functional form was chosen as a simple means to force $Q_{i-i}$ to zero for one ion where no heating exists, while allowing $Q_{i-i}$ to converge to the analytical form in the large-number limit[20]. The power of the $N_i/N_k$ term in the exponent was determined by inspection of the comparison between the analytical model and simulated dynamics shown in Fig. 1c. No attempts were made to refine the model since it also gives a satisfactory answer for the ion steady state, as shown in Fig. 1d. Here $N_k$ is the ion number at which the ion configuration transitions from a one-dimensional chain to a three-dimensional crystal, representing the conditions under which the sample can be adequately described by the parameters of a plasma, which we determine, for our experimental configuration, to be 22 ions.

Last, the analytical approximation for $\log \Lambda$ given in ref. 20) is extended to low temperatures by including a temperature-dependent function $\zeta(T_{sec}) = (1 + 10e^{-T_{sec}/T_S})$. This modification manifests itself in a roughly tenfold increase in the heating rate for ion systems with temperatures less than $T_s < 10$ mK, as suggested empirically in ref. 20.

**MD simulation.** The sympathetic cooling process is simulated with molecular-dynamnics software ProtoMol. Ion trajectories are calculated by numerically integrating Newton's equation

$$m_i \frac{d^2 \mathbf{r}_i}{dt^2} = \mathbf{F}_{LQT}(\mathbf{r}_i) + \frac{Q^2}{4\pi\varepsilon_0}\sum_{j \neq i}\frac{\mathbf{r}_i - \mathbf{r}_j}{|\mathbf{r}_i - \mathbf{r}_j|^3} + \mathbf{F}_{i-n} \quad (3)$$

using a leapfrog algorithm. The force due to the LQT is given by $\mathbf{F}_{LQT} = -\nabla V_{LQT}$, with

$$V_{LQT}(\mathbf{r}) = \frac{x^2 - y^2}{r_0^2}V_{r.f.}\cos(\Omega t) + \kappa\frac{z^2 - \frac{1}{2}(x^2 + y^2)}{z_0^2}V_{ec} \quad (4)$$

and $r_0$, $z_0$, $V_{rf}$, $V_{ec}$, $\kappa$, and $\Omega$ are field radius, axial length from trap centre to end-cap, recombination frequency (r.f.) voltage, end-cap voltage, geometrical factor and trap frequency, respectively. The integration algorithm chooses a step size $\Delta t$ that is much smaller than the r.f. period $2\pi/\Omega$.

The ion-neutral collision term, $\mathbf{F}_{i-n}$, is approximated as an instantaneous event which happens at an average rate of $\Gamma_{i-n}$ and changes ion's velocity $\mathbf{v}_i$ elastically according to,

$$\mathbf{v}_i' = \frac{m_i\mathbf{v}_i + m_n\mathbf{v}_n}{m_i + m_n} + \frac{m_n}{m_i + m_n}\mathcal{R}(\mathbf{v}_i - \mathbf{v}_n) \quad (5)$$

while leaving the ion position $\mathbf{r}_i$ unchanged. $\mathcal{R}$ is a random rotation matrix into all solid angles for each collision event, which serves as a good approximation of the real elastic cross-section[11].

Cold initialization for the idealized simulation is performed by mimicking the laser-cooling of the ions before immersion by including a velocity-dependent damping term in equation (3), which sets the cold initial temperature to $T_c^i \leq 1$ mK. Hot initialization is performed by simulating the white noise heating of the ions by the inclusion of random voltages between $-100$ and $+100$ mV on two laterally paired LQT electrodes that update at each time step. This simulated electrical white noise sets the ion temperature to $50 \text{ K} \leq T_h^i \leq 500$ K. Both of these initializing terms are turned off once immersion begins. The simulations for the ideal case are performed 10 times for each ion number to account for the statistical nature of the collisional process.

Initialization for the realistic simulation is modified slightly from the ideal simulation to account for a small amount of electrical noise present in the hybrid trap. To this end, an additional white noise term with an amplitude of $\sim 150$ μV is included during initialization and remains on during immersion to simulate this noise on the trap electrodes. In addition, the realistic simulation uses a Gaussian cold-atom cloud with a $1 - \sigma$ radius of 0.46 mm and peak density of $2.3 \times 10^9$ cm$^{-3}$. The simulations for the realistic case are performed 30 times for each ion number, again to account for the statistical nature of the collisional process.

The unstable calcium ion background is created primarily by the direct ionization of calcium atoms in the $4p^1P_1$ state by a 369 nm laser, which we include to dissociate calcium dimer ions produced in the laser-cooled buffer gas. We measure this photodissociation cross-section to be $\sim 1 \times 10^{-18}$ cm$^2$ and estimate a loading rate of 6,000 ions per second. The loading is simulated by creating an calcium ion at a density-weighted position within the buffer gas every 167 μs. The simulation predicts the unstable ion to be present for roughly two r.f. trap cycles before it escapes the trap.

**Experimental details.** The hybrid atom–ion trap discussed here consists of a co-located calcium magneto-optical trap (MOT) and segmented LQT. The MOT is composed of six orthogonal, Doppler-cooling lasers that drive the $4p^1P_1 \leftarrow 4s^2 {}^1S_0$ cooling transition in $^{40}$Ca at 423 nm. A laser at 672 nm repumps calcium atoms that decay into the $3d^1D_2$ state back to the ground state via the excited $5p^1P_1$ state, closing the cooling cycle. These lasers each coincide near the null of a quadrupole magnetic field with a 60 G cm$^{-1}$ gradient at the trap centre. Neutral calcium atoms are introduced by a heated getter unit, subsequently decelerated by far-detuned ($4\,\Gamma$ and $10\,\Gamma$) cooling lasers, and ultimately loaded into the MOT. Imaging of the atoms is performed by two near-orthogonal electron multiplying charge coupled device (EMCCD) cameras using a 25 ms exposure time to observe 423 nm fluorescence.

The segmented LQT has a field radius $r_0 = 6.85$ mm and electrode radius $r_e = 4.50$ mm. The $\Omega = (2\pi)680$ kHz trapping potential is applied asymmetrically with one diagonal pair of electrodes having an r.f. amplitude $V_{rf} = 173$ V and the other pair at r.f. ground, leading to a Mathieu-$q$ of 0.28. Axial confinement is provided by biasing the outer electrodes by $V_{ec} = 4$ V. Central electrodes are also biased to compensate for excess micromotion of the ions. Barium ions are loaded into the trap by the ablation of a solid BaCl$_2$ target mounted beneath the LQT. The ions are then preinitialized by ramping the voltages applied to the central electrodes to iteratively lower the trap depth, thereby decreasing the number of trapped ions from large thermal clouds to chains of ions. Throughout this preinitialization, axial and radial 493 nm cooling ($6p\ ^2P_{1/2} \leftarrow 6s\ ^2S_{1/2}$) and 650 nm repump ($6p\ ^2P_{1/2} \leftarrow 5d\ ^2D_{3/2}$) lasers cool the ions to $< 100$ mK. Imaging of the ions is performed by a single reentrant EMCCD camera using a 50 ms exposure time with a gain of 200 and 10 Hz frame rate to observe 650 nm fluorescence.

Heating of the ions for the hot measurements is performed by applying white noise to a central LQT electrode for 0.2 s before immersion. The spectral density of the white noise applied to the central LQT electrode at the secular frequency of the ions ($\omega_s \sim 2\pi \times 55$ kHz) is measured to be $1.0 \times 10^{-6}$ (V m$^{-1}$)$^2$ Hz$^{-1}$ at the location of the trapped ions. The heating of the ions is confirmed by the disappearance of the imaged ion fluorescence during this heating. Laser-cooling of the ions ceases and loading of the calcium MOT begins 100 ms before the white noise heating ends to ensure the ions remain hot at the beginning of immersion. Without the white noise heating, the power of any residual electronic noise on the central LQT electrode at the secular frequency of the ions is measured to be below the noise floor of our spectrum analyser.

Experimentally, immersion times $> 5$ s are not feasible due to background collisions, which result in both heating and ion loss. In addition, long immersion times can lead to chemical reactions between the trapped ions and neutral species which result in further ion loss and the formation of molecular ions.

**Data analysis.** Cuts made to the experimental data set include hot trials that are not heated sufficiently, cold trials that are not adequately laser-cooled before immersion, trials with a final ion number less than the initial ion number and trials with more than one 'dark' ion (for short immersion times these are typically barium ions which fall into the $^2$D$_{5/2}$ state that is not addressed by the laser-cooling scheme). 'Sufficiently heated' is implied by the complete disappearance of ion fluorescence during secular excitation, which implies temperatures $> 50$ K. 'Sufficiently cooled' is implied the perseverance of crisp, resolved ion fluorescence before immersion, which implies temperatures $< 10$ mK. Trials that have a final ion number less than the initial ion number are likely due to a number of experimental realities, including background gas collisions/reactions and heating due to, for example, background Ca$^+$ ions and ion–ion heating above the trap depth.

**Data availability.** All relevant data are available from the authors on request.

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

## Acknowledgements

This work was supported by National Science Foundation (PHY-1205311) and Army Research Office (W911NF-15-1-0121 and W911NF-14-1-0378) grants.

## Author contributions

E.R.H. and K.C. conceived the theoretical concept. E.R.H., S.J.S. and A.J.D. conceived the experiment and measurement protocol. C.S. and S.J.S. built the apparatus. S.J.S., A.J.D. and P.P. acquired and analysed all of the data. K.C. developed the MD software and A.J.D. performed all of the simulations presented. S.J.S. wrote the manuscript and S.J.S., A.J.D. and C.S. prepared all of the figures with input from all authors.

## Additional information

**Competing financial interests:** The authors declare no competing financial interests.

