## [Peer Review file · Nature Communications]

Editor's Comments:

“I would ask you to revise your title as our article style requires no punctuation in titles.”

Our Response: We have changed the title to “Blue-sky bifurcation of ion energies and the limits of neutral-gas sympathetic cooling of trapped ions.”

Referee #1:

1. “I believe that it would be important to clearly point out for which of the listed applications the effects studied in the current manuscripts are relevant.”

Our Reply: We agree with the Referee and have added the following to the conclusion of the paper in order to emphasize and contextualize the relevancy of this result:

“Outside of hybrid trapping, these limitations may impact low-temperature chemical studies that use neutral gases to set reaction energies and the development of mass spectrometers, which use cold neutral gases to collisionally cool analyte ions for increased mass resolution. This work may also revise the viability of using cold neutral gases *alone* as an initialization step for molecular ion qubits in future quantum computation architecture. Finally, the ion-ion heating rate applied in the present work should also be applicable to studies of bistability in laser-cooled trapped ions.” (Page 6, Paragraph 1).

2. “Laser cooling of trapped ions gives also rise to bistable behaviour. It would be important to compare the instabilities studied in the paper with the ones found in laser cooling of a few ions. Is there something to be learned from the new experiments? Is the model developed applicable to the laser-cooling experiments? Do the instabilities studied in the manuscript differ from the ones found when trying to laser-cool trapped ions in rf-traps, or are they of a similar nature?”

The bistable behavior of laser-cooled ions arises from a variety of factors including the relative axial and radial secular frequencies, the velocity dependence of the Doppler cooling force, photon recoil in some cases, and the ion-ion heating rate. The last of these is, of course, one of

the main drivers of the bifurcation in the present work. Thus, there is certainly some overlap in the underlying physics and our work explaining this heating could be used in a description of laser cooling bistability. We previously did not comment on this relationship because a proper description of laser cooling bi-stability requires inclusion of several topics significantly beyond our present work. Nonetheless, as the Referee has pointed out, it would be wise to draw attention to the fact that the heating rate described here can be used in those studies, so that other workers can benefit from the description. Therefore, we have added to the conclusion:

“Finally, the ion-ion heating rate applied in the present work should also be applicable to studies of bistability in laser-cooled trapped ions.”(Page 6, paragraph 1)

3. “In the paragraph following equation (2), an ‘empirically motivated’ function $\xi(N)$ is introduced, whose form is given in the appendix. I would like to know which criteria were applied when choosing this form.”

Our Response: We agree with the Referee that it’s important to inform the reader the motivation of the form used. We have expanded the relevant section in the Methods so that it now reads: “For small numbers of ions, ρ is poorly defined and it becomes impractical to derive an analytical form for the ion-ion heating rate due to its dependence on the actual ion trajectories. We therefore modify the heating rate by the inclusion of the function $\xi(N) = 1 - e^{-N/N_c}$. This functional form was chosen as a simple means to force Q to zero for one ion where no heating exists, while allowing Q to converge to the analytical form in the large-number limit [20]. The power of the N/N_c term in the exponent was determined by inspection of the comparison between the analytical model and simulated dynamics shown in Fig. 1c. No attempts were made to refine the model since it also gives a satisfactory answer for the ion steady state, as shown in Fig. 1d. Here, N_c is the ion number at which the ion configuration transitions from a one-dimensional chain to a three-dimensional crystal, representing the conditions under which the sample can be adequately described by the parameters of a plasma, which we determine, for our experimental configuration, to be 22 ions,” (Page 6, Paragraph 4).

4. “I found some of the introduction a bit hard to follow. For example, the authors refer for the justification of (2) to ref. 20; however, reading ref. 20, it is not straightforward to find the required explanation. It would be important to make the manuscript a bit more self-contained.”

Our Response: We apologize to the Referees for the tendency to divert the text away from self-consistency. We understand that this overly frequent referencing of our supporting work tends to weaken the focus of this paper. To maintain the focus of the introduction (specifically in the explanation of Equation 1), we have included the following text:

“This heating rate is simply the product of an effective ion-ion collision rate Γ_{i-j} and the fractional increase in ion energy per collision ϵ , which vanishes for static traps[21], but is nonzero for LQTs. Γ_{i-j} is traditionally approximated using the Chandrasekhar-Spitzer plasma self-collision rate [22], while the determination of ϵ is given in Ref. ([20]),” (Page 2, Paragraph

1).

5. “I found it confusing to see that in Fig. 2 the recoiling time seems to be shorter for the ,hot/cold control' experiments, where the sympathetic cooling is absent. Is this a general feature?”

Our Response: The Referee points out a good place in the text for us to clarify some of the subtleties of the cooling/heating dynamics when the MOT is on versus when the MOT is off. To elucidate these nuances we have included the following text in reference to Fig. 2:

“As seen in Fig. 2, the recoiling times for the hot and cold measurements are each larger than the respective hot and cold control measurements. For the cold measurements, this is an expected and general trend due to laser-cooling resulting in initial temperatures less than $T_n \sim 10$ mK. For the hot measurements in Regimes II and III, this is due to the fact that the hot initialization aims

only to prepare the ions above ν_1 , which is less than ν_1 ,” (Page 4, Paragraph 1).
And later:

“This Ca^+ background adds a small but non-negligible heat load to the ions which can also help explain why hot and cold measurements are each larger than the respective hot and cold control measurements,” (Page 4, Paragraph 2).

6. “In Table I, the ion temperature for buffer gas cooling with 4He is given for a buffer gas density of $10^{14}/\text{cm}^3$; in the text, a density of $10^{13}/\text{cm}^3$ is mentioned.”

Our Response: We thank the Referee to pointing out the error and have fixed the error to the correct value of 10^{14} cm^{-3} , (Page 5, Paragraph 3).

7. “To support the conclusion, it would be useful if another column could be added to the table listing the ion temperatures that are expected when the ions are preinitialized to low temperatures. In this case, it would of course also be necessary to point out how to achieve this in the first place.”

Our Response: We agree with the Referee and have added the two extra columns conveying the unstable (bifurcation) temperature and the “maintain” temperature value. Additionally, we have emphasized the following text to clearly explain to the reader how the ions could be preinitialized below the bifurcation temperature:

“In most cases, this issue these issues can be circumvented either by directly laser-cooling the trapped ions or by sympathetically cooling the ions with co-trapped laser-cooled ions, which can maintain (or simply initialize, in the case of cooling with Na and Rb gases) ions at sub-millikelvin temperatures,” (Page 5, Paragraph 4).

Referee #2

1. Improve the discussion of the Analytical Model in the methods sections to make it more self-contained.

Our Response: Again, we apologize to the Referees from the lack of self-containment. To correct this, we have heavily modified the description of the Analytical Model in the Methods to now read:

“The kinetics factor λ suppresses the cooling rate which accounts for the less-than-unit cooling efficiency per collision. As referred to in the text, this inefficiency is a consequence of ongoing micromotion interruption, which results in some collisions leading to cooling and others to heating. As λ approaches zero, fewer collisions result in cooling. The micromotion interruption also leads to the cooling asymptote $\overline{T_n}$ being greater than the reservoir temperature T_n . The micromotion interruption is a function of trap parameters and atom masses, but generally worsens for increasing Mathieu- q and m_n/m_i ratio.

“The heating rate defined in Equation 2 is developed in Ref. ([20]). Rather than model the secular temperature T_{sec} , here we approximate the total ion temperature T such that $T_{sec} = \eta T$. The ratio of secular to total temperature η is a function of trap parameters and decreases as micromotion increases relative to the secular motion. In practice, it is convenient to treat micromotion as an additional degree of freedom. Here, the axially-confining field is much weaker than the radially-confining field which leads to micromotion being relevant in only two dimensions. Then, by equipartition, $\eta \sim 3/5$. Exact calculation of η using Ref. ([20]) yields 0.50,” (Page 6, Paragraph 2-3).

Referee #3

1. One point that should be clarified is the regime of validity for the analytic approximations for the ion-atom cooling and ion-ion heating rates. For example, recent work (arXiv:1512.04197) demonstrated the breakdown of the cooling rate approximation in Equation 1 under certain conditions, suggesting that perhaps the limitations described in the current work can be circumvented in some circumstances. While the molecular dynamics simulations and experimental data support the use of the models in the regime studied here, a brief discussion of the conditions for the analytic approximation to hold would be appropriate.

Our Response: We agree with the Referee that stating the assumptions is important to the reader. To clarify under which conditions the analytical model is appropriate we have added the following text:

“This model assumes an isotropic cold-atom reservoir, isotropic momentum transfer differential cross-section, and does not incorporate short-range dynamics due to the C_4 atom-ion interaction,” (Page 2, Paragraph 0).

This means, as the Referee no doubt already knows, that the departure of the data in arXiv:1512.04197 from e.g. our Eq 1 is because in that work the buffer gas is smaller than the sampled trap volume. In this case, the value of ϵ will change dramatically as heating collisions tend to only happen at large distances from the trap center, which is obviously precluded in the

arrangement of arXiv:1512.04197.

2. Hot trials that are not heated sufficiently: earlier discussion seems to indicate that this determination is made based on whether the ion fluorescence disappears before immersion. Is this indeed how such trials are selected out?

3. Cold trials that are not adequately laser-cooled: again, are these trials identified by the absence of ion fluorescence before immersion, or based on some other criterion?

4. Trials with a final ion number less than the initial ion number: it seems that one likely explanation for these events is ions acquiring sufficient kinetic energy to escape the trap. Is the frequency of these events consistent with this interpretation, e.g. with ions being lost more frequently under conditions that otherwise lead to longer recooling times? Or is there a mechanism other than heating thought to dominate ion losses?

Our Response (2,3,4): We agree with the Referee that elucidation of the process for which data is cut is necessary. In response we have added the following text to the Data Analysis section of the Methods:

“‘Sufficiently heated’ is implied by the complete disappearance of ion fluorescence during secular excitation, which implies temperatures > 50 K. ‘Sufficiently cooled’ is implied the perseverance of crisp, resolved ion fluorescence prior to immersion, which implies temperatures < 10 mK. Trials which have a final ion number less than the initial ion number are likely due to a number of experimental realities, including background gas collisions/reactions and heating due to e.g. background Ca^+ ions and ion-ion heating above the trap depth,” (Page 7, Paragraph 8).

Reviewer #1 (Remarks to the Author):

The revised version of the manuscript addresses the points I mentioned in my first review in a satisfactory manner. I therefore recommend publication of the paper in Nat. Comm. in its present form.

Reviewer #3 (Remarks to the Author):

I am pleased with the changes the authors have made, which strengthen the manuscript and completely address the minor concerns I expressed. I believe the paper is now suitable for publication.